**Perspective**

# Antarctic science operations must account for climate change and extreme environmental events
Martin Siegert [1] ✉, Thomas J. Bracegirdle[2], Peter Convey [2,3], Katharine Hendry [2], Caroline Holmes [2] & Kevin A. Hughes [2]

Extreme environmental events (EEEs) in Antarctica and the Southern Ocean are occurring with growing frequency and severity, and will become more pronounced under continued global warming. Antarctic research relies on safely deploying personnel and equipment into remote locations. Preparing and undertaking science is, thus, dependent on conditions at sea and on ice. Here, we discuss impacts on scientific operations from EEEs and assess the vulnerability of research stations, the operability of airfields and non-prepared landing sites and reliability of vessel access. We show that each will face challenges to offering service levels that have been the hallmark of Antarctic scientific discovery. To mitigate, fieldwork must be supplemented where possible and appropriate by autonomous systems. As we build up to the 5th International Polar Year (2032-3), understanding how Antarctic research can flourish in the face of EEEs – including consideration of duration, magnitude, location and spatial gradient – will become a critical theme.

There is increasing evidence that anthropogenic activities such as fossil fuel burning and widespread deforestation, and consequential global heating of 1.4 °C to date[1], have led to the increased occurrence and severity of extreme environmental events (EEEs)—sudden incidents of substantial and/or unexpected deviation from regular conditions[2,3]. In Antarctica, EEEs have become recognised across a variety of realms and disciplines (terrestrial, cryospheric, atmospheric, oceanographic)[4], with some occurring measurably at increasing rates and intensities. Antarctica is now subject to record sea-ice lows, heatwaves, storms and summer surface melting[4], with formal attribution to climate change in at least one case[5], and records likely to be broken further up to and beyond the 1.5 °C Paris Climate agreement's suggested 'safe' limit for planetary health[6,7].

Forming a universal definition of Antarctic EEEs, which fits all circumstances, is challenging. Antarctica has an intrinsically extreme environment, which is already subject to magnified change relative to that seen in the rest of the planet due to fossil fuel burning and resultant global heating. On top of this backdrop, incidents occur that can take the form of short shocks, irreversible step changes or persistent changes acting across many different aspects or features of the environment. We offer an inclusive approach to EEEs by considering the effects of such incidents, which can be individual, collective or cascading, on essential operations and logistics in Antarctica and the Southern Ocean.

One key domain that has experienced recent extremes is Antarctic sea ice cover. Following record high levels between 2013 and 2015, by 2023 Antarctic sea ice has reduced considerably, recording persistent lows with respect to the month, with an overall record minimum in February 2023[4]. Since then, winter months (June–October) and late summer (February) have remained at near-record low levels—lower than anything observed prior to 2022 in the satellite era (1978–present), and very likely the lowest in the 21st century. Furthermore, in February 2025, the world experienced a record *global* sea ice low[8]. This marks 10 years of very low sea ice levels in Antarctica since 2015, with only temporary recovery to near-average around 2020 (Fig. 1), indicating a structural change to persistent lower sea ice conditions[9].

Periods of low summer sea ice extent have wide-ranging negative impacts on the Antarctic physical system; they amplify warming within the Southern Ocean, likely as an outcome of heat uptake, and are associated with increased levels of iceberg calving[10]. They also drive mostly negative impacts on marine biology and ecosystems, through shifts in the timings and extent of nutrient supply, light availability, and habitats[10]. For example, open water modifies the seasonality and levels of plankton blooms, which will, in turn, affect the food chain. The loss of sea ice will also affect prey species, such as krill, that rely on it for refuge, and higher predators, such as penguins, seals and certain whales, which need sea ice for hauling out, foraging and breeding.

[1]University of Exeter, Penryn, UK. [2]British Antarctic Survey, Cambridge, UK. [3]Department of Zoology, University of Johannesburg, Johannesburg, South Africa. ✉e-mail: m.siegert@exeter.ac.uk

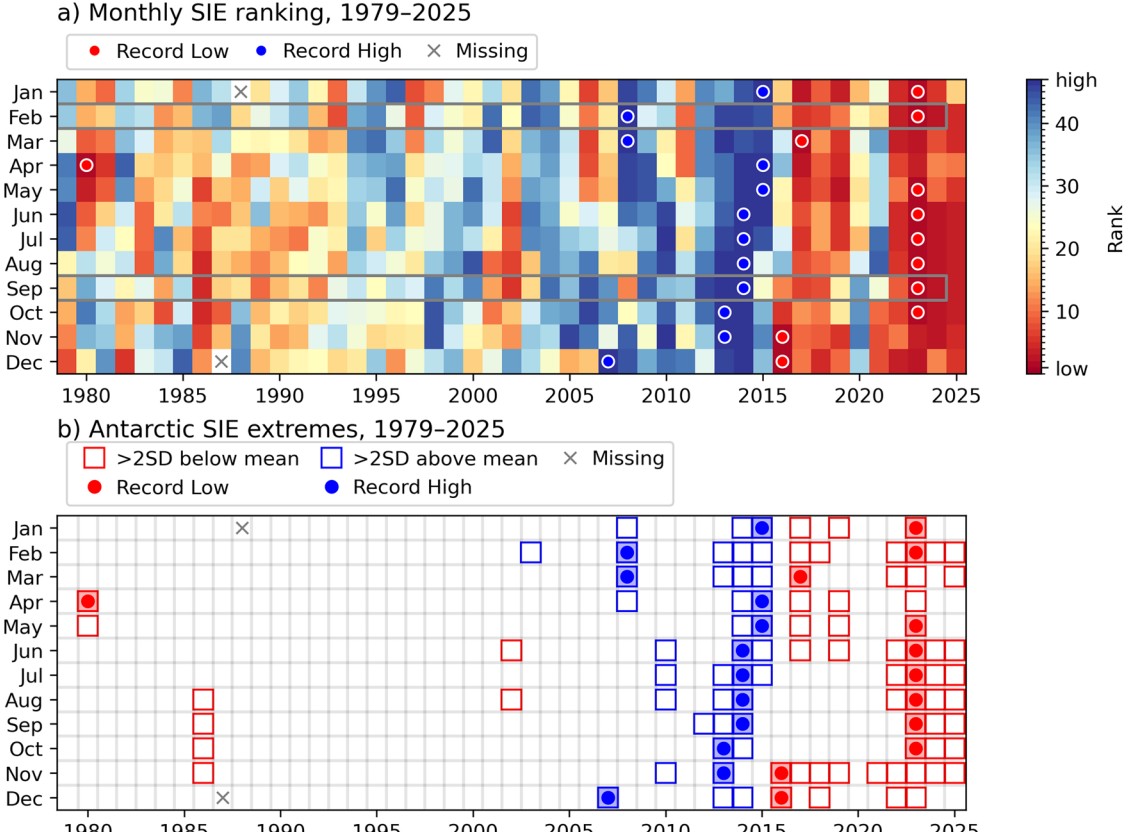

**Fig. 1 | Monthly Antarctic sea ice variations since 1979. a** Ranking of continental sea ice extent (circles denote record highs in yellow and lows in white), with maximal areas in blue and minimal in red. **b** Sea ice extents noted as squares when deviating from the long-term mean (1981–2010) by 2 Standard Deviations (blues high, reds low) with records denoted as circles. Data source: Monthly Sea Ice Extent (SIE) from the NSIDC Sea Ice Index v4 (Fetterer et al.[70]).

On the continent, as a consequence of high surface air temperatures, there has been enhanced pooling of surface water, with measurable spread onto the grounded East Antarctic Ice Sheet[11]. Indeed 30% of the ponding reported in Antarctica is now seen over grounded ice. Floating ice surface melting covers the remainder, with recent record levels on, for example, the George VI Ice Shelf between Alexander Island and the Antarctic Peninsula mainland[12].

EEEs are becoming common, expected and serious disrupters of what has historically been considered the stable deep polar marine environmental system of Antarctica. As global warming continues, with its magnification at around 2× in the Antarctic[13], the expectation is for more EEEs to be experienced in the decades ahead. Changes in extreme conditions do not in general linearly follow changes in background mean climate and are unlikely to be uniform across the continent and across different types of events[14], with particular events having location-specific and type-specific outcomes and gradients of influence. However, much of the continent is likely to be affected in some way.

Here, based on our experiences and knowledge (primarily with, but not restricted to, UK operations), we discuss how EEEs will lead to impacts on science and logistics in Antarctica and the Southern Ocean, and how—if at all—such impacts can be mitigated. We consider how scientific experimentation may be adversely affected through reassessments of risk, operations and policy. We note that governmental National Antarctic Programmes operate under a wide variety of circumstances, and each will need to determine how EEEs affect activities and plans in ways highlighted by, but not restricted to, our considerations.

## Evidence of disruption to scientific planning and operations

Science in Antarctica has until now largely escaped serious disruption due to extreme events – with the obvious exception of COVID-19[15]—but that is not to say it has been unaffected or that increased effects are unlikely in future.

## Risks to station buildings

The first period of substantial continent-wide Antarctic station construction occurred in the build up to the International Geophysical Year (1957/58), when some of the largest Antarctic stations were built. In the following decades further stations were established by nations often associated with their efforts to attain consultative status under the Antarctic Treaty, which gives them voting rights in Antarctic governance. Today, more than 80 stations are operated by c. 31 Antarctic Treaty Parties, with around 50% located on the Antarctic Peninsula and Scotia Arc South Shetland Islands and South Orkney Islands[16]. Most are pragmatically located on climatically less extreme and accessible coastal and ice-free locations, facilitating resupply by sea, although a small number are located inland and constructed on permanent ice. National governments are increasingly taking climate change and extreme events into consideration as they construct new or redevelop existing stations. In a 2018 survey of the Council of Managers of National Antarctic Programs (COMNAP) member countries, 73% were undertaking station modernisation works and, of them, 22% cited climate change as the main reason[17]. More recently, COMNAP, an ideal forum for sharing information on building design and for disseminating lessons that can be learnt from experience both in Antarctica and from the Arctic, has provided an assessment of infrastructures in a changing Antarctica, where national Antarctic programmes were encouraged to continue to share their technical and practical expertise[18].

Buildings in Antarctica have always needed to cope with the extreme conditions that exist there. In recent decades, however, the impacts of warmer, wetter and more humid conditions mean buildings (bar those on the ice sheet plateaux) must now be able to deal with flooding events, heavy rainfall or snow melt, large snow loads, and be constructed of materials that are resistant to wet rot. Under EEEs this problem will be exacerbated. For example, coastal stations may even need to be protected from the possibility of lightning strikes—an issue that has seldom been considered in Antarctica.

Designers and managers must solve the problem of stabilising buildings located on thawing permafrost[19] and select locations that are not vulnerable to sea level rise impacts (although isostatic rebound and reduced gravitational influences may reduce the risk in some locations)[20]. Increasing temperatures have reduced the stability of the ice-piers constructed annually at McMurdo Station, that has now been replaced by a barge pier[21,22]. At some stations, extreme weather conditions have accelerated the degradation of building materials such as insulation and sealants[23]. Shoreline erosion resulting from more frequent storms and reduced ice cover has resulted in modernisation work and construction of coastal defences at a number of stations, including beach reinforcement with eco-friendly blocks to reduce the erosion[18,23]. Furthermore, altered precipitation patterns and hydrography may mean stations can no longer rely upon freshwater lakes and streams for their water supplies[24]. Climate change and changing EEEs may result in the emergence of once-buried station buildings and abandoned waste dumps, which can result in further dispersal of waste and/or mobilisation of pollutants[25–27].

## Conditions of airstrips

Access to Antarctica and to deep field locations often requires the use of aircraft. While some airstrips are prepared using a crushed rock surface on solid ground, others are on either floating or grounded ice. Remote operations relying on ski-equipped aircraft (generally smaller types such as the de Havilland Twin Otter, Basler BT-67 or Dornier 228) commonly land on unprepared and unsurveyed locations. EEEs can affect each of these runway types through snow accumulation, strong (cross) winds and poor visibility, all of which can reduce the operational window. At the Rothera Research Station rock airstrip, the formation of a solid ice layer on the runway surface has delayed planned aircraft movements by up to 6 weeks, with serious consequences for programme operations. The British Antarctic Survey is currently undertaking an environmental impact assessment on use of de-icer on the runway, but this will need to adhere to the Antarctic Treaty's strict environmental protection guidelines (relating to the release of chemicals into the environment) before use. On ice, McMurdo Station has previously relied on an operational runway on sea ice; however, although still commissioned, it has remained unused since the 2010s due to limitations in sea ice stability[28]. Flights were moved to its Pegasus airstrip—an established compacted snow runway on a nearby ice shelf—which itself ceased operations in 2016 due to excessive summer melting. Flights are now more regular at Williams Field, which currently has a more robust season-long schedule. At Wilkins Aerodrome, serving Casey Station, another compacted snow runway, this time on grounded ice, excessive summer melt has reduced the number of flights possible. Initially planned at one flight per week (around 20 per season), this number is seldom reached. In fact, from 2011 the runway has been temporarily unusable each season due to melting. Summer surface snow melt and exposure of crevasses have halted use of the snow airstrip near Teniente Carvajal station at the southern edge of the Fuchs Ice Piedmont, Adelaide Island, necessitating the search for possible new locations of crevasse-free runways at higher elevations[29]. While on-ice runways are essential for Antarctic research, they are vulnerable to extreme weather and climate conditions and, as temperatures and extreme events increase in intensity and frequency, they are likely to suffer further vulnerability and restrictions. With increasing EEEs, an inability to predict accurately snow and ice mechanics at temperatures around melting point, as well as greater variability in snow temperature and thickness, may increasingly comprising runway stability and use[30].

## Surface water and field operations

Away from established runways and stations, summer snow melt leading to slush accumulation on ice/snow surfaces provides a further challenge to remote air-supported deep field operations, as well as to ground travel relying on vehicles such as snowmobiles. At the extreme, considerable 'ponding' of water can occur on ice shelves, as seen in recent years on Larsen A, B and C, in ways associated with sudden shelf collapse[31]. Extensive melting on George VI Ice Shelf[12] presents an added danger through being connected by an extensive network of sometimes deep and fast-flowing drainage systems, including sink holes draining through the ice. Extensive summer snow melt has increasingly temporally limited operational aircraft landing access for operators supporting field parties in the South Shetland Islands and northern Antarctic Peninsula since at least the late 1990s (e.g. the closure of the Damoy Point transit facility near Port Lockroy in 1993).

In terms of operational ability to support the required regular maintenance of key long-term scientific activities, a pertinent example is provided by the British Antarctic Survey's inability to land aircraft in order to access Mars Oasis on southern Alexander Island since the mid-2010s. The site has since 2000 been the location of one of very few year-round long-term biological microenvironmental monitoring stations anywhere in Antarctica, and the furthest south location at low altitude in the Antarctic Peninsula[32]. However, it has suffered from the combination of increasingly early and rapid snow melt on the adjacent Mars Glacier surface, and the increasing extent of ponding and melt stream activity on the nearby part of George VI Ice Shelf. Consequently, Mars Oasis had not been successfully visited since the 2015/16 season, with the decision being taken in 2025 to abandon and clear the site (achieved in November 2025), and intention to redeploy at a location close to Fossil Bluff in a subsequent season. However, Fossil Bluff itself, while a long-established British Antarctic Survey 'forward operational support facility', also relies on an unprepared snow skiway that faces periods of extensive slush formation. Furthermore, access to its small station building located on the lateral moraine on eastern Alexander Island has faced periods of severe obstruction by the formation of extensive and deep seasonal melt lakes along the edge of the moraine. The disruption to, and ultimately loss of, such a substantial long-term monitoring programme through rapidly increasing operational constraints caused by climate change and EEEs constitutes a serious scientific programme impact. More work is needed to understand the development of, and gradients in, slush in order to plan field operations in future.

The switch of summer precipitation from snow to rain in the northern and central Antarctic Peninsula and Scotia Arc archipelagos compound these melt-related challenges. Further impacts are provided by thaw events occurring with more frequency in winters. Thaw events can lead to hard but potentially brittle ice layers forming within the snowpack and/or on the underlying ground surface. At the extreme, this can lead to the replacement of snow cover with an impenetrable ice layer that can strongly affect the thermal regime of the underlying soil/habitats and restrict/stop oxygen transfer. In the Arctic, these processes are compounded by increasing frequency and intensity of winter 'rain-on-snow' (RoS) events, which can lead to rapid and extensive loss of snow cover and formation of thick ground surface ice layers[33]. As well as potentially wide-ranging and negative consequences for soil ecosystems and their contained and dependent organisms, such events also generate travel safety risks for human communities and operations. While such winter RoS events are yet to be documented in the Antarctic Peninsula region, winter thaws and ice layer formation have long been documented at Signy Island (South Orkney Islands)[34] and are likely increasing in frequency.

## Ships and ocean access

Changes in sea ice thickness, duration and extent are having substantial impacts upon Antarctic shipping activities, affecting national Antarctic programmes and the tourism and fishing industries. Sea ice decline may facilitate access to Antarctic research stations and visitor sites for more of the year, as has been modelled for tourist landings in the western Antarctic Peninsula under different climate scenarios[35]. Growing levels of Antarctic ship traffic may increase environmental impacts, including through the introduction of marine non-native species[36,37], additional ship emissions[38,39] and black carbon[40], and the potential for shipping accidents, especially from non-ice strengthened vessels that may encounter sea ice conditions with which they are not equipped to deal. Sea ice decline may facilitate access of ship-borne tourists and researchers to new remote locations with associated increases in human footprint[41,42] and environmental impacts such as

trampling, pollution, wildlife disturbance and the introduction or dispersal of non-native species and wildlife pathogens[43,44].

Changes in sea ice can also impact Antarctic ship operations. For example, low sea ice has recently facilitated operations in winter with, for example, the UK's RRS *Sir David Attenborough* and Chile's *Almirante Viel* having undertaken Antarctic research station visits and oceanographic and marine biological research during the early and late 2025 winter period, respectively. Conversely, late sea ice breakup can delay ship access to research stations[45]. These impacts depend strongly on local sea ice conditions, and while recent declines have been marked by relative homogeneity[46], sea ice anomalies often have strong compensating regional signals, with even recent low years marked by temporary, locally extensive sea ice conditions, such as in the western Amundsen Sea in July 2023[47].

Furthermore, unexpected changes in ice shelf and multi-year sea ice extent have resulted in the loss of depoted fuel and equipment, as well as hindering station resupply[48,49]. Sea ice extent also affects where and when fishing occurs. For example, in many years sea ice has restricted krill fishery operations around the South Orkney Islands, leading to relocation of the fishing fleet[50]. Conscious of the potential environmental risks associated with sea ice decline, policymakers within the Antarctic Treaty System (ATS) have tasked the Subsidiary Group on Climate Change Response to consider the issue further[51].

## Assurance on health and safety
EEEs have substantial consequences for the health and safety of those engaging in Antarctic science and operations, and indeed on tourism and fishing. Increasingly unpredictable sea ice conditions mean that the risks associated with travel over sea ice, including by over-ice vehicles, have increased and the practice is often now prohibited for safety reasons. Access by shipping to new uncharted waters as a consequence of sea ice decline and ice shelf collapse increases the risk of shipping incidents and associated pollution events. Furthermore, national Antarctic programmes may have reduced confidence in the reliability of what were routinely used off-load sites on ice shelves; greater monitoring efforts are now made to provide assurance that sites are sufficiently safe with ship times alongside being kept to a minimum. Changing temperature regimes, particularly during earlier periods of the summer season, may result in increased risks of rock/ice fall and avalanches at sites that were previously considered stable and predictable. Variable precipitation patterns and increasing air temperatures may reduce snow fall and increase ice melt, respectively, resulting in the elimination of once safe over-ice travel routes due to crevasse risk. This has occurred at Rothera Research Station where the route linking the coastal research station with a snow skiway on the adjacent inland ice piedmont has become temporarily (possibly permanently) impassable due to the opening of crevasses, and the ice ramp adjacent to the station becoming too steep and icy for vehicles to use safely for large parts of the year[52,53]. This also has important implications for station personnel morale as the skiway provides a means to 'get away' from the station, and its closure represents a loss of recreation options, especially for those who are on station for several months or more.

## The risks and mitigations of ensuring deep field access
Understanding key glacial processes requires access to extremely remote locations. Numerous examples of deep-field sites have delivered outstanding scientific advances, but such science relies on the ability to safely deploy, operate and recover staff (including in unplanned situations such as requirement for medevac), equipment and fuel. Such work can be considered as being at the end of the logistics chain, involving ships, aircraft and stations, all of which – as we note above – have been and are likely to be increasingly impacted by future EEEs. Planning deep-field activities is complex under normal circumstances but, with the challenges of extreme weather and the (in)ability to supply specific sites, further complexity is added. There is no doubt that science will continue to demand access to remote field locations in future, but planning such research is likely to be impacted by EEEs in ways that may make such work ever more challenging to accomplish.

Some mitigations exist, however. For example, use of longer-range aircraft and drones, rather than types that require deep-field fuel caches, could de-risk airborne observations. Reduced human presence in general, if the science can be achieved through remote means, is likely to be encouraged in future. Use of remotely operated and autonomous devices is likely to become an increasingly major theme of investigation, where these are practically applicable. When it is necessary to deploy personnel, international collaboration for field access is a further means to reduce uncertainty, through increased routes and services, although bringing other challenging practical constraints, such as compatibility of operational and health and safety procedures, and the prevalence of post-COVID protectionism through lack of confidence in the ability to rely on other operators[15].

## Prioritising research into extreme events
In 2014, the Scientific Committee on Antarctic Research (SCAR) commissioned a 'horizon scanning' exercise, to understand the most important scientific questions relating to Antarctica and the Southern Ocean that require answers within 20 years[54]. Aligned with this, COMNAP assessed the logistical needs for such work, effectively describing how to achieve a 20-year international scientific plan[55]. Reference to EEEs featured broadly within these international exercises, with a firm appreciation of the need for research into how and why rapid changes in Antarctica affect the world. By 2019, a quarter of the way through the 20-year timeframe of the horizon scan, work on extreme events was beginning to be covered—a notable transformation from 5 years earlier—with the role of atmospheric rivers on surface ice sheet mass balance receiving particular attention, along with extreme sea-ice conditions and the impacts of marine heatwaves on marine biodiversity[56]. There was also an appreciation of the likely increase in such events in future. Hence, the scientific community was becoming attuned to the risks and importance of extreme events in Antarctica.

We are now more than halfway through the horizon scan's 20 years—a period that has seen multiple Antarctic EEEs, often record breaking, and increased prioritisation of research into them. Further access to remote and challenging locations is likely to be in demand to fulfil SCAR's revised ambition, especially for the upcoming 5th International Polar Year (IPY, 2032-3), which will be at the tail end of the SCAR 20-year 'scan'. This IPY offers an opportunity for an internationally-connected research programme into Antarctic EEEs. Such a programme could focus on understanding: (1) how and why Antarctic EEEs occur, (2) how EEEs relate to anthropogenic climate change, including attribution studies, (3) the inter-connectivity of EEEs across systems, (4) the impacts of EEEs on human infrastructure (e.g. research stations, science flights, tourism, heritage sites[57], shipping and fisheries), (5) how Antarctic EEEs affect global environmental processes (e.g. sea level rise, weather systems, oceanic flow, heat transfer), (6) how Antarctic EEEs lead to global impacts to infrastructure (e.g. ports and cities, marine navigation, non-Antarctic fisheries), and (7) how to mitigate their impact on science using drones and other autonomous systems.

## International cooperation and the Antarctic Treaty
For most of the last ~65 years, Antarctic science, and its governance through the ATS, has been driven predominantly by the twelve original signatories to the Treaty (Argentina, Australia, Belgium, Chile, France, Japan, New Zealand, Norway, South Africa, the Soviet Union (now Russia), the UK and the USA) and collaborations within and outwith this group, especially with various EU member states. In the last decade, however, a noticeable shift in Antarctic research has occurred with a decline in scientific outputs from these nations alongside a rapid rise in research from China, which has nearly tripled its publications in this time and, indeed, has now overtaken the USA as the 'lead' Antarctic output nation in terms of publication number[58]. In a continent devoted to peace and science and where the demonstration of substantial scientific research activity is necessary to participate in Antarctic governance, and with 58 signatories to the Treaty, a shift in scientific activity and change in levels of academic output may have broad implications[59].

These could include an increasing role by some Parties in the provision of research to respond to the information and knowledge needs of the bodies of the Antarctic Treaty system (such as the Antarctic Treaty Consultative Meeting, the Committee for Environmental Protection and the Commission for the Conservation of Antarctic Marine Living Resources (CCAMLR)), as well as a changing relationship between Parties operating key infrastructure, such as airstrips, and those seeking to use these facilitate to deliver their field research activities.

This shift in Antarctic research power, and research investment, has been rapid and comes alongside noticeable changes to the Antarctic environment that are exemplified by the rise in extreme events. While these international changes in research power are not necessarily unwelcome or alarming, they do point to a future in which the Antarctic environment, and our ability and desire to understand it, will likely be different in the years ahead compared with previously. This presents an important new paradigm that Antarctic nations must acknowledge and work with, with implications for revised national and international arrangements and governance.

In the recent UK national security strategy[60], a fear was expressed that agreements on how to mitigate the effects of climate change may be diminished if multilateral institutions decline in influence. The UK's formal position is, thus, to uphold the ATS as a necessary and best way to seek and maintain agreement internationally. A specific key commitment is to strengthen climate research, including on tipping points and global impacts of Antarctic change[61]. As climate change leads to intensification of EEEs, the ATS must adapt to and recognise the challenges that emerge.

Regarding how the ATS is positioning itself for EEEs, it is the role of the Committee for Environmental Protection to provide advice to the Antarctic Treaty Consultative Meeting. Responding to recommendations in the SCAR Antarctic Climate Change and the Environment report[62,63], the Committee for Environmental Protection developed the Climate Change Response Work Programme, which included the issue of 'climate change impact to the built (human) environment resulting in impacts on natural and heritage values'[59,64–66].

In support of the ATS awareness of EEEs, SCAR offers advice on the latest scientific findings through international panels and reports. For example, the SCAR Antarctic Climate Change and the Environment Decadal Synopsis[67], recognised that work in Antarctica, built over past decades of relative environmental stability, will be challenging to continue in future due to loss of floating and land ice, and because of EEEs.

Recognising the nuanced relationships between the ATS, COMNAP and SCAR[68], a possible way forward is for the ATS to first formulate precisely and carefully articulate to SCAR and COMNAP their knowledge needs, alongside a plan for how to use this knowledge in decision-making. SCAR, working with COMNAP, can then decide how best it can respond within the limited resources available to them in terms of committees and programmes, in order to offer the ATS sufficient recommendations that allow international agreements and planning.

## Adapting to the rise of Antarctic extreme events

Antarctic research has benefited enormously from the structures, systems and expertise that have made such work possible throughout the era of the Antarctic Treaty. This way of working was initially largely developed during a period of relative environmental stability and positive engagement in international cooperation through the ATS. However, as a consequence of global warming and the rise of EEEs that are likely linked to this, the way in which we consider Antarctic research in coming years may differ from the past. EEEs are, first surprisingly[69] but now rather expectedly, breaking records in terms of heatwaves, sea ice loss, snowfall and rain episodes, all of which can adversely affect the planning and undertaking of fieldwork and operations, including the important consideration of health and safety. There are many examples of station rebuilding in line with the expectation of EEEs, which can contribute to mitigating the problems raised, as well as scientific plans that may help reduce the human presence in the deep field through increased use of remote sensing where this is appropriate. SCAR's 2014 assessment of the most urgent scientific questions that required

answers within 20 years is now over halfway through and will come to an end around the time of the upcoming 5th IPY (2032-33). Assessment of EEEs loomed large in that exercise, and is highly likely to be even more important for the IPY and thereafter. Antarctic EEEs will impact future science ambitions and so their investigation and ways to mitigate their impacts on operations is sure to become a critical focus. The way in which we work in Antarctica will always be influenced by our past experience and expertise, but future work will also require appreciation of changes to the Antarctic environment that currently have seldom, or never, been experienced.

## Data availability

NSIDC Sea Ice Index v4 Sea Ice Extents (Fetterer et al.[70]) used to produce Fig. 1 are available from https://nsidc.org/sea-ice-today/sea-ice-tools by downloading the spreadsheet Sea_Ice_Index_Monthly_Data_with_-Statistics_G02135_v4.0.xlsx. Note that the in-spreadsheet rankings will change as future data is added.

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

## Acknowledgements
This paper benefitted from discussions held at the Royal Society of London during a Hooke Discussion Meeting on 29–30 October 2025, led by KH. We thank three referees for offering constructive advice that improved the paper considerably. PC acknowledges intellectual discussions with colleagues in the Cape Horn International Centre (CHIC), Puerto Williams, Chile, and the Millennium Institute- Biodiversity of Antarctic and Sub-Antarctic Ecosystems (BASE), Santiago, Chile.

## Author contributions
M.S. oversaw the compilation of the paper. T.J.B. provided text on Antarctic extreme weather events. P.C. contributed text on terrestrial ecology and deep field operations. K.H. provided text on ocean-related extreme events and operations. C.H. compiled Fig. 1 and provided text on sea ice extremes. K.A.H. provided advice on national and international governance regarding Antarctic protection and field operations. All authors contributed to revising and editing the paper.

## Funding
K.H. acknowledges support from the Royal Society for a Hooke Discussion Meeting HDMR\100078 and from NERC National Capability grant PRESCIENT (NE/Y006178/1). T.J.B. acknowledges support from NERC Highlight Topic ExtAnt (Drivers and Impacts of Extreme Weather Events in Antarctica) (NE/Y503307/1). C.H. acknowledges support from NERC Highlight topic DEFIANT (NE/W004747/1). P.C. and K.A.H. are supported by NERC core funding to the British Antarctic Survey 'Biodiversity, Evolution and Adaptation Team' and Environment Office, respectively.

## Competing interests
The authors declare no competing interests
