## [Transparent Peer Review file · Communications Earth & Environment]

Antarctic science operations must account for climate change and extreme environmental events

Corresponding Author: Professor Martin Siegert

Version 0:

Decision Letter:

Dear Professor Siegert,

Your manuscript titled "Antarctic science operations must account for climate change and extreme environmental events" has now been seen by 3 reviewers, and we include their comments at the end of this message. They find your work of interest, but some important points are raised. We are interested in the possibility of publishing your study in Communications Earth & Environment, but would like to consider your responses to these concerns and assess a revised manuscript before we make a final decision on publication.

We therefore invite you to revise and resubmit your manuscript, along with a point-by-point response that takes into account the points raised. Please highlight all changes in the manuscript text file.

Please submit your point-by-point responses as a separate file, distinct from your cover letter where you can add responses to the Editors' comments that you do not want to be made available to the reviewers. Word files are preferred. We recommend that any figures, tables or graphs that are included in the response to reviewers are also included in the main article or Supplementary Information.

Please use the following link to submit your revised manuscript, point-by-point response to the referees' comments (which should be in a separate document to any cover letter), a tracked-changes version of the manuscript (as a PDF file) and the completed checklist:

Link Redacted

We hope to receive your revised paper within six weeks; please let us know if you aren't able to submit it within this time so that we can discuss how best to proceed. If we don't hear from you, and the revision process takes significantly longer, we may close your file. In this event, we will still be happy to reconsider your paper at a later date, as long as nothing similar has been accepted for publication at Communications Earth & Environment or published elsewhere in the meantime.

Please do not hesitate to contact us if you have any questions or would like to discuss these revisions further. We look forward to seeing the revised manuscript and thank you for the opportunity to review your work.

Best regards,

Nicola Colombo, PhD
Associate Editor, Communications Earth & Environment
Consulting Editor, Communications Sustainability

EDITORIAL POLICIES AND FORMATTING

- Behavioural and social science
- Ecological, evolutionary & environmental sciences
- Life sciences

Furthermore, please align your manuscript with our format requirements, which are summarized on the following checklist: [Communications Earth & Environment formatting checklist](https://www.nature.com/documents/commsj-phys-style-formatting-checklist-article.pdf)

and also in our style and formatting guide [Communications Earth & Environment formatting guide](https://www.nature.com/documents/commsj-phys-style-formatting-guide-accept.pdf) .

*** DATA: Communications Earth & Environment endorses the principles of the Enabling FAIR data project (<http://www.copdess.org/enabling-fair-data-project/>). We ask authors to make the data that support their conclusions available in permanent, publically accessible data repositories. (Please contact the editor if you are unable to make your data available).

All Communications Earth & Environment manuscripts must include a section titled "Data Availability" at the end of the Methods section or main text (if no Methods). More information on this policy, is available at <http://www.nature.com/authors/policies/data/data-availability-statements-data-citations.pdf>.

If a community resource is unavailable, data can be submitted to generalist repositories such as [figshare](https://figshare.com/) or [Dryad Digital Repository](http://datadryad.org/). Please provide a unique identifier for the data (for example a DOI or a permanent URL) in the data availability statement, if possible. If the repository does not provide identifiers, we encourage authors to supply the search terms that will return the data. For data that have been obtained from publically available sources, please provide a URL and the specific data product name in the data availability statement. Data with a DOI should be further cited in the methods reference section.

REVIEWER COMMENTS:

Reviewer #1 (Remarks to the Author):

The manuscript titled "Antarctic science operations must account for climate change and extreme environmental events" by Siegert and others makes interesting and timely points about the threat to Antarctic infrastructure from ongoing climate change and, more broadly, the need for investment in Antarctic research. I believe it will be a valued contribution by many groups, both scientists and managers, and as such I encourage publication. With that said, several points could use some simple clarification so that both experts and nonexperts alike can make use of the final product.

Lines 36-53: This paragraph does not mention the record high sea-ice extent that happened just a ~decade ago and more recently on a local scale in some areas. Both figures 1 and 2 show this in some way though. It deserves mention here otherwise seems overlooked in the text despite being in figures.

Line 58: Stating mostly negative biological consequences without any explanation of causes or details or variability just

opens the door to questions and doubts. It seems like a throwaway line but rather than adding detail, it is adding doubt. Cut, or else significantly expand.

§2.1: Many places in the Arctic have had to adapt building strategies in areas of permafrost or for potential flooding due to expanded melt seasons. Are there any lessons to be learned?

Lines 104-112: The explanation about limited sea-level rise in Antarctica due to GIA and gravitational change is clear. However, a few sentences later we read about sea-level rise impacting coastal stations, and it doesn't seem to take into account the information that preceded. Is this risk only in certain areas? Or, is it quite minor? How much shoreline erosion in Antarctica is truly due to sea-level rise, and how much is just increased storminess, or reduced protection from sea ice? Since this paragraph starts with saying the impact is reduced in Antarctica compared to elsewhere, when the effect is then claimed it needs some clarifications.

Lines 109-110: Yes, materials degrade quickly in extreme environments like Antarctica. However, that is not part of climate change or increasing extreme events, it is just the background of extreme climate there. Be sure that statements about general decay or aging are separated out from those statements that discuss threats in a changing climate. Isn't the paper cited here just about general extreme conditions?

§2.4: The discussion of health and safety would better follow the section on deep field and come as a wrap of all the discussions of physical infrastructure. As it is, it seems to float here in the middle of section 2.

§2.5: What about expanded access for research by standard research vessels, not just icebreakers or ice-strengthened vessels? Many countries are expanding their Antarctic research fleet by adding small, unstrengthened vessels. This deserves mention here, including the potential for increased risk should unexpected sea-ice conditions be met.

Line 316: There is certainly a correlation between journal profile and quality of a published manuscript. However, it is not 1:1. The report cited simply quantifies the number of papers each country has put out, and the number in high profile journals. They do not assess the quality of the publications, which is the claim being made in this sentence. Rewrite and do not make a claim greater than in the original.

Figure 1: It might be helpful to have panels (a) and (b) line up so that the x-axes of each are the same length. As long as two panels cover the same range of years, it is natural for the reader to glance back and forth and expect the years to align.

Figure 2: This figure is a nice idea. However, as it is, it is just pretty at best, and confusing at worst. What determines where on the continent each inset sits? Am I to interpret there are lots of runways at 0° Longitude? Or that we should only worry about safe deployment of people in the Ross Sea? Additionally, the text is difficult to read. I am not sure of the best solution to this but possibly split the map of the continent to a different figure and label the stations. For the "infographic," the inset panels could be lined up in grid with marine/terrestrial on one axis and timescale of threat or change on the other. Or, infrastructure v. purely physical (rain on snow, surface melt) on one. Anyway, something that might put these panels into an order that means something, rather than just scattered around the continent in a semi-arbitrary way.

I look forward to seeing the final version of this contribution.

Sincerely, Julia Wellner

Reviewer #2 (Remarks to the Author):

The manuscript focuses on how extreme events are impacting on human activities in Antarctica. The topic is extremely interesting, and it covers a wide range of activities related to scientific infrastructure. The general framework is largely acceptable but some points should be better shaped:

1- Authors associate impacts on facilities to extreme events severity and frequency: Figure 1 highlights a tipping point in 2016 when a diffuse heating is evidenced. Can they separate the impact of extreme events from the overall warming in the continent? What is the weight of EEEs compared to the regional climate change?

2- The analysis claims about a diffused impact of EEEs, but are there spatial gradients in Antarctica? Coastal to inland for sure...

3- Station buildings: the description is qualitative but are there spatial distribution of the presented case studies? Where are required coastal defenses? Is there a relation with the age of stations?

4- Airstrips: a wider analysis could highlight additional issues. The Pegasus airstrip documented problems in the '90s not associated with EEEs but to blue ice degradation (this is the reason why it was turned to white ice). Could it be wind-albedo feedback the cause of degradation instead of EEEs? What about sea ice runways? Examples are ok, but different case studies could be included in the presented discussion

5- Surface water: slush is an emerging phenomena, are there spatial gradients affecting occurrence and stations?

6- Safety: the presented example supports the claiming of a diffused issue? Are there spatial gradients for the presented observations?

7- Sea ice and shipping: are there spatial gradients for the presented occurrences?

In conclusion the core of the final conclusion is focusing the attention on EEEs, for sure exceptional considering a climatological record, but will EEEs anomalous in the next years if we are documenting a tipping point in the last ten years?

Reviewer #3 (Remarks to the Author):

This Perspective addresses a timely and important topic. The authors are well placed to discuss how extreme environmental events (EEEs) are disrupting Antarctic scientific operations. The manuscript contains several relevant examples, but in its current form it requires major revision before it meets the standard expected of a Communications Earth & Environment Perspective.

A CE&E Perspective is expected to present a clear, forward-looking argument rather than mainly compile known problems. As written, the manuscript reads more like a structured synthesis than a paper built around a distinct and actionable central message. The main claim — that EEEs are increasingly affecting Antarctic operations and that adaptation is needed — is reasonable and generally well supported, but it is not, on its own, sufficiently sharp for this format. The paper would be stronger if the authors committed more clearly to a specific recommendation the community has not yet fully adopted, such as formalised logistics sharing, explicit deprioritisation of some deep-field activities during high-risk periods, or dedicated 5th IPY investment in autonomous observing capability.

Major concerns:

The first major issue is the definition of EEEs. The manuscript defines them as sudden and unexpected events, yet much of the evidence discussed, especially the persistent low sea-ice conditions since 2015, reflects sustained environmental change rather than discrete extreme events. This conceptual tension should be addressed more explicitly.

The second major issue is geographic imbalance. Most operational examples are drawn from BAS-linked facilities, with McMurdo and Wilkins Aerodrome serving only as limited counterpoints. Major national programmes such as Concordia, Neumayer, Syowa, Jang Bogo, and Zhongshan are absent. This is particularly noticeable because Section 4 raises China's growing Antarctic research role while not incorporating Chinese operational experience into the broader discussion. The authors should either broaden the evidence base or explain why the current examples are sufficiently representative. The mitigation discussion also needs further development. The abstract promises discussion of how impacts can be mitigated, but this remains underdeveloped in the main text. Autonomous underwater vehicles are mentioned in the abstract but not meaningfully discussed in the body, and land-based autonomous platforms are effectively absent. More attention should also be given to the practical constraints on adaptation, including cost, infrastructure hardening, and logistical redundancy.

Finally, Section 4 raises an interesting governance point but does not fully develop it. The observation that China has overtaken the USA in Antarctic research output could have important implications, but these are only briefly noted and not analysed in depth. If this argument is retained, its implications for burden-sharing, infrastructure dependence, and Antarctic governance should be discussed more directly.

Minor concerns:

- There are also some reference and presentation issues that should be corrected. The EPA ice pier reference appears to need renumbering.
- Reference [51], a The Conversation article, is not a sufficiently strong source for the claim about China's leadership in Antarctic research output; a peer-reviewed bibliometric source would be more appropriate.
- The 2018 COMNAP survey data cited in Section 2.1 are now dated and should be updated if more recent figures are available.
- The abstract and body should also be aligned more carefully with respect to mitigation using autonomous platforms.

** Visit Nature Portfolio's author and referees' website at www.nature.com/authors for information about policies, services and author benefits**

Communications Earth & Environment is committed to improving transparency in authorship. As part of our efforts in this direction, we are now requesting that all authors identified as 'corresponding author' create and link their Open Researcher and Contributor Identifier (ORCID) with their account on the Manuscript Tracking System prior to acceptance. ORCID helps the scientific community achieve unambiguous attribution of all scholarly contributions. You can create and link your ORCID from the home page of the Manuscript Tracking System by clicking on 'Modify my Springer Nature account' and following the instructions in the link below. Please also inform all co-authors that they can add their ORCIDs to their accounts and that they must do so prior to acceptance.
<https://www.springernature.com/gp/researchers/orcid/orcid-for-nature-research>

If you experience problems in linking your ORCID, please contact the [Platform](http://platformsupport.nature.com/)

Support Helpdesk.

Version 1:

Decision Letter:

Dear Professor Siegert,

Your manuscript titled "Antarctic science operations must account for climate change and extreme environmental events" has now been seen by our reviewers. In light of their advice we are delighted to say that we are happy, in principle, to publish a suitably revised version in Communications Earth & Environment provided you clearly articulate limitations in representativeness, particularly as they relate to Antarctic programmes run by other countries, for example, China.

We therefore invite you to revise your paper one last time. At the same time we ask that you edit your manuscript to comply with our format requirements and to maximise the accessibility and therefore the impact of your work.

EDITORIAL REQUESTS:

*****Please take care to match our formatting and policy requirements. We will check revised manuscript and return manuscripts that do not comply. Such requests will lead to delays. *****

SUBMISSION INFORMATION:

OPEN ACCESS:

Communications Earth & Environment is a fully open access journal. Articles are made freely accessible on publication. For further information about article processing charges, open access funding, and advice and support from Nature Portfolio, please visit <https://www.nature.com/commsenv/open-access>

Link Redacted

Best regards,

Nicola Colombo, PhD
Associate Editor, Communications Earth & Environment
Consulting Editor, Communications Sustainability

**** Visit Nature Portfolio's author and referees' website at www.nature.com/authors for information about policies, services and author benefits****

Response to referees' comments

"Antarctic science operations must account for climate change and extreme environmental events"

Dear Nicola,

Many thanks for forwarding three referees' comments on the above paper. We found all three reviews generally supportive of the paper, with a variety of recommendations and concerns that we address below.

As you'll see, we have made a number of changes to the paper, and attach a new version with the changes noted in track mode, as well as a clean version.

Details of how we respond to referees are made in red.

Kind regards,

Martin Siegert (on behalf of the authors).

Reviewer #1 (Julia Wellner) (Remarks to the Author):

The manuscript titled "Antarctic science operations must account for climate change and extreme environmental events" by Siegert and others makes interesting and timely points about the threat to Antarctic infrastructure from ongoing climate change and, more broadly, the need for investment in Antarctic research. I believe it will be a valued contribution by many groups, both scientists and managers, and as such I encourage publication. With that said, several points could use some simple clarification so that both experts and nonexperts alike can make use of the final product.

We thank the referee for their support and for offering constructive ideas for how the paper could be improved.

Lines 36-53: This paragraph does not mention the record high sea-ice extent that happened just a ~decade ago and more recently on a local scale in some areas. Both figures 1 and 2 show this in some way though. It deserves mention here otherwise seems overlooked in the text despite being in figures.

We have modified the text to read as follows: "Following record high levels of sea ice between 2013 and 2015, by 2023 Antarctic sea ice formed considerable and persistent lows with respect to the month, with an overall record minimum in February 2023 [4]."

Line 58: Stating mostly negative biological consequences without any explanation of causes or details or variability just opens the door to questions and doubts. It seems like a throwaway line but rather than adding detail, it is adding doubt. Cut, or else significantly expand.

We do explain how the loss of sea ice can affect the physical environment and now offer similar thoughts on why it affects the biological system.

New text is as follows: “For example, open water will modify the seasonality and levels of plankton blooms, which will in turn affect the food chain. The loss of sea ice will also affect prey species, such as krill, that rely on it for refuge, and higher predators, such as penguins and seals, which need sea ice for hauling out and breeding.”

§2.1: Many places in the Arctic have had to adapt building strategies in areas of permafrost or for potential flooding due to expanded melt seasons. Are there any lessons to be learned?

COMNAP would be the place where such lessons are learnt and we have now indicated this is the case in the following new text: “COMNAP is an ideal forum for sharing information on building designs, and for disseminating lessons that can be learnt from experience both in Antarctica and from the Arctic, where building on permafrost, for example, has been a challenge for many years.”

Lines 104-112: The explanation about limited sea-level rise in Antarctica due to GIA and gravitational change is clear. However, a few sentences later we read about sea-level rise impacting coastal stations, and it doesn't seem to take into account the information that preceded. Is this risk only in certain areas? Or, is it quite minor? How much shoreline erosion in Antarctica is truly due to sea-level rise, and how much is just increased storminess, or reduced protection from sea ice? Since this paragraph starts with saying the impact is reduced in Antarctica compared to elsewhere, when the effect is then claimed it needs some clarifications.

We have edited the text to take out the influence of sea level rise, as shoreline erosion is more obviously affected by storms and loss of sea ice over the coming years.

Lines 109-110: Yes, materials degrade quickly in extreme environments like Antarctica. However, that is not part of climate change or increasing extreme events, it is just the background of extreme climate there. Be sure that statements about general decay or aging are separated out from those statements that discuss threats in a changing climate. Isn't the paper cited here just about general extreme conditions?

Many thanks for this comment. We agree of course that buildings will be subject to a triple issue of extreme conditions, climate change and EEEs. It's challenging to completely separate the effects, but we have made the text clearer that EEEs will exacerbate the problems. The new text is as follows:

“Buildings in Antarctica traditionally need to cope with the extreme conditions that exist there. In recent decades, however, the impacts of warmer, wetter and more humid conditions mean buildings (bar those on the ice sheet plateaux) must now be able to deal with flooding events, heavy rainfall or snow melt, large snow loads, and be constructed of materials that are resistant to wet rot. Under EEEs this problem will be exacerbated. For example, coastal stations may even need to be protected from the possibility of lightning strikes – an issue that has seldom been considered in Antarctica.”

§2.4: The discussion of health and safety would better follow the section on deep field and come as a wrap of all the discussions of physical infrastructure. As it is, it seems to float here in the middle of section 2.

We have moved section 2.4 as suggested.

§2.5: What about expanded access for research by standard research vessels, not just icebreakers or ice-strengthened vessels? Many countries are expanding their Antarctic research fleet by adding

small, unstrengthened vessels. This deserves mention here, including the potential for increased risk should unexpected sea-ice conditions be met.

We have added to the section on this point as follows: “...and the potential for shipping accidents especially from non-ice strengthened vessels that may encounter sea ice conditions with which they are not equipped to deal”

Line 316: There is certainly a correlation between journal profile and quality of a published manuscript. However, it is not 1:1. The report cited simply quantifies the number of papers each country has put out, and the number in high profile journals. They do not assess the quality of the publications, which is the claim being made in this sentence. Rewrite and do not make a claim greater than in the original.

This has been edited as suggested. We don't refer to quality but do maintain a 'shift' in research power as that is undeniable in recent years.

Figure 1: It might be helpful to have panels (a) and (b) line up so that the x-axes of each are the same length. As long as two panels cover the same range of years, it is natural for the reader to glance back and forth and expect the years to align.

This is a good point and can be done during typesetting.

Figure 2: This figure is a nice idea. However, as it is, it is just pretty at best, and confusing at worst. What determines where on the continent each inset sits? Am I to interpret there are lots of runways at 0° Longitude? Or that we should only worry about safe deployment of people in the Ross Sea? Additionally, the text is difficult to read. I am not sure of the best solution to this but possibly split the map of the continent to a different figure and label the stations. For the “infographic,” the inset panels could be lined up in grid with marine/terrestrial on one axis and timescale of threat or change on the other. Or, infrastructure v. purely physical (rain on snow, surface melt) on one. Anyway, something that might put these panels into an order that means something, rather than just scattered around the continent in a semi-arbitrary way.

We have tried a number of alternatives to this infographic. It's intended to quickly portray the variety of influences that extreme events can have on the continent, rather than a scientific diagram. We'd prefer to keep it roughly the same, as a means to quickly exemplify the problems in Antarctica. We have added new commentary to each of the issues highlighted. We would willingly discuss further modifications with the editorial team.

I look forward to seeing the final version of this contribution.

Again, many thanks for your support.

Reviewer #2:

The manuscript focuses on how extreme events are impacting on human activities in Antarctica. The topic is extremely interesting, and it covers a wide range of activities related to scientific infrastructure. The general framework is largely acceptable but some points should be better shaped.

We thank the referee for their support and for offering ways in which the paper may be improved. Quite a few of the responses concern ‘**gradients**’ in extreme events. This is an important point to make, but not one that we can confidently resolve in all places here. More work is certainly needed. We make sure to highlight this as an important consideration. For example, we have added the following text to the abstract. “But, as we build up to the 5th International Polar Year (2032-3), understanding how Antarctic research can flourish in the face of EEEs – in terms of length, magnitude, location **and gradient** –will become an increasingly challenging theme.”

1- Authors associate impacts on facilities to extreme events severity and frequency: Figure 1 highlights a tipping point in 2016 when a diffuse heating is evidenced. Can they separate the impact of extreme events from the overall warming in the continent? What is the weight of EEEs compared to the regional climate change?

It would in principle be possible to do this, but would be highly depending on the specific impact being assessed and the specific definition of extreme event being used. For example, a tipping point results in conditions which are extreme relative to the previous state, but not the new state. We are not aware of any formal analysis making this comparison. Important context to this is that recent research has shown that for Antarctica (as found generally elsewhere) the tails of the distribution (i.e. extremes) do not in general follow background mean change. For example, extremes in heavy precipitation increase more quickly than the mean due to increased moisture capacity of the atmosphere and also depend on shifting storm tracks. Temperature extremes in the polar regions also change at different rates from the mean at many locations as surface conditions (in particular shifts from snow/ice cover to ice free land/ocean) affect variability in temperature.

We have added a new paragraph at the top of the paper to help this issue:

“Forming a universal definition of Antarctic EEEs, which fits all circumstances, is challenging. Antarctica has an intrinsically extreme environment, which is already subject to magnified change relative to that seen in the rest of the planet due to fossil fuel burning and resultant global heating. On top of this backdrop, incidents occur that can take the form of short shocks, irreversible step changes or persistent changes acting across many different aspects or features of the environment. We offer an inclusive approach to EEEs, by considering the effects of such incidents, which can be individual, collective or cascading, on essential operations and logistics in Antarctica and the Southern Ocean.”

We have also now added the latter point to the second-last paragraph of the Introduction, with the following:

“Changes in extreme conditions do not in general linearly follow changes in background mean climate and are unlikely to be uniform across the continent and across different types of events [14], with particular events having location-specific and type-specific outcomes and gradients of influence. However, much of the continent is likely to be affected in some way.”

2- The analysis claims about a diffused impact of EEEs, but are there spatial gradients in Antarctica? Coastal to inland for sure...

We have indicated that this may be the case, to promote new work. New text is as follows: “Such change is unlikely to be uniform across the continent, with particular events having location-specific

and type-specific outcomes and gradients of influence. However, much of the continent is likely to be affected in some way.”

3- Station buildings: the description is qualitative but are there spatial distribution of the presented case studies? Where are required coastal defenses? Is there a relation with the age of stations?

As far as we are aware, sufficient information is not publicly available to allow a spatial analysis of the impacts of extreme events on station buildings. We have highlighted the examples of building medication provided in the recent paper to the Committee for Environmental Protection by the Council of Managers of National Antarctic Programs (COMNAP, 2025).

The sentence on coastal defences has been amended to read ‘Shoreline erosion resulting from more frequent storms and reduced ice cover has resulted in modernisation work and construction of coastal defences at a number of stations, including reinforcement with eco-friendly blocks to reduce the erosion’

4- Airstrips: a wider analysis could highlight additional issues. The Pegasus airstrip documented problems in the ‘90s not associated with EEs but to blue ice degradation (this is the reason why it was turned to white ice). Could it be wind-albedo feedback the cause of degradation instead of EEs? What about sea ice runways? Examples are ok, but different case studies could be included in the presented discussion

We make the point that airstrips on snow and ice will be impacted by rising temperatures as well as by extreme events. The referee is correct to make the point that Pegasus was moved over 10 years ago. It is not the moving that is critical, but the usability of it hereon due to warming and extremes.

We have added a further example about ice runways as highlighted by the reviewer: ‘Summer surface snow melt and exposure of crevasses have halted use of the ice airstrip near Teniente Carvajal station at the southern edge of the Fuchs Ice Piedmont, Adelaide Island, necessitating the search for possible new locations of crevasse-free runways at higher elevations’.

Also: ‘With increasing extreme events, an inability to predict accurately snow and ice mechanics at temperatures around melting point, as well as greater variability in snow temperature and thickness may increasingly comprising runway stability and use.’

5- Surface water: slush is an emerging phenomena, are there spatial gradients affecting occurrence and stations?

This is a good point, and there are likely to be spatial gradients in surface water properties, but we are unaware of documented evidence. We have made the point to encourage work in this area. New text as follows: “More work is needed to understand the development of, and gradients in, slush in order to plan field operations in future.”

6- Safety: the presented example supports the claiming of a diffused issue? Are there spatial gradients for the presented observations?

We use examples that predominantly reflect the experience of personnel operating through the logistics of the British Antarctic Survey, which operates predominantly in the Antarctic Peninsula region. The examples provided reflect changes in safety for personnel working across the BAS operational footprint, with extreme events impacting those in marine and terrestrial environments, including areas of permanent ice. To undertake a spatial analysis of Health and Safety incidents across BAS operation, and looking for any spatial gradients, would be likely be possible by examining data in the BAS incident reporting system. However, this would be a substantial piece of work, far beyond the aims of the current paper. Nevertheless, we shall investigate this suggestion further and we are grateful to the review for suggesting this as a useful line of research.

7- Sea ice and shipping: are there spatial gradients for the presented occurrences?

To our knowledge, few studies have looked at spatial gradients in terms of sea ice extent impacts on shipping. However, we have now mentioned a study looking at predicted future tourist landings in the Antarctic Peninsula region, which expand southward as sea ice declines: “Sea ice decline may facilitate access to Antarctic research stations and visitor sites for more of the year, as has been modelled for tourist landings in the western Antarctic Peninsula under different climate scenarios [Lee, 2019].”

The spatial gradients typical in sea ice anomalies are worthy of mention here, and text has been added accordingly in the context of impacts on ship operations: “These impacts depend strongly on local sea ice conditions, and while recent declines have been marked by relative homogeneity [Schroeter et al, 2023], sea ice anomalies often have strong compensating regional signals, with even recent low years marked by temporary, locally extensive sea ice conditions, such as in the western Amundsen Sea in July 2023 [Gilbert and Holmes, 2023].”

The core of the final conclusion is focusing the attention on EEEs, for sure exceptional considering a climatological record, but will EEEs anomalous in the next years if we are documenting a tipping point in the last ten years?

Our position is that EEEs should be expected to grow in significance in coming years. Whilst of importance, our commentary doesn't consider tipping points, which would require a new piece of work. No changes are made.

Reviewer #3:

This Perspective addresses a timely and important topic. The authors are well placed to discuss how extreme environmental events (EEEs) are disrupting Antarctic scientific operations. The manuscript contains several relevant examples, but in its current form it requires major revision before it meets the standard expected of a Communications Earth & Environment Perspective.

A CE&E Perspective is expected to present a clear, forward-looking argument rather than mainly compile known problems. As written, the manuscript reads more like a structured synthesis than a paper built around a distinct and actionable central message. The main claim — that EEEs are

increasingly affecting Antarctic operations and that adaptation is needed — is reasonable and generally well supported, but it is not, on its own, sufficiently sharp for this format.

Honing this perspective is important to have the impact desired by all. We thank the referee for their general support and note their opinion on its present level of effectiveness. The comment about forming explicit recommendations is made well. While the issue of extreme events in Antarctica will be familiar to many who work on it, the value of publishing in *Communications Earth & Environment* means that it will reach non-traditional audiences, and thus needs to be relevant broadly.

It must be for an editor to decide whether the balance is right. This piece was invited after a meeting held at the Royal Society, based on a plenary talk on which the paper is based. Deviation from the broad boundaries of that would not have been appropriate at this stage.

However, we do want to improve and maximise the impact of the paper, and offer responses below to specific comments.

The paper would be stronger if the authors committed more clearly to a specific recommendation the community has not yet fully adopted, such as formalised logistics sharing, explicit deprioritisation of some deep-field activities during high-risk periods, or dedicated 5th IPY investment in autonomous observing capability.

We presently state: “a possible way forward is for the ATS to first formulate precisely, and carefully articulate to SCAR and COMNAP their knowledge needs, alongside a plan for how to use this knowledge in decision making. SCAR, working with COMNAP, can then decide how best it can respond within the limited resources available to them in terms of committees and programmes, in order to offer the ATS sufficient recommendations that allow international agreements and planning.”

We also note that “The IPY offers an opportunity for an internationally-connected research programme into Antarctic EEEs. Such a programme could focus on understanding: (1) how and why Antarctic EEEs occur, (2) how EEEs relate to anthropogenic climate change, including attribution studies, (3) the inter-connectivity of EEEs across systems, (4) the impacts of EEEs on human infrastructure (e.g., research stations, science flights, tourism, heritage sites [56], shipping and fisheries), (5) how Antarctic EEEs affect global environmental processes (e.g., sea level rise, weather systems, oceanic flow and heat transfer), and (6) how Antarctic EEEs lead to global impacts to infrastructure (e.g. ports and cities, marine navigation, non-Antarctic fisheries).”

We have now added a new line, as recommended, on the use of drones. “...and (7) how to mitigate their impact on science using drones and other autonomous systems.”

This, we note, is quite diplomatic language. We’d like to take everyone along in considering how to build knowledge in this area, rather than dictate what should be done.

Major concerns:

The first major issue is the definition of EEEs. The manuscript defines them as sudden and unexpected events, yet much of the evidence discussed, especially the persistent low sea-ice conditions since 2015, reflects sustained environmental change rather than discrete extreme events. This conceptual tension should be addressed more explicitly.

We refer to EEEs as “sudden incidents of substantial and unexpected deviation from regular conditions”. We agree that the definition is qualitative, but that is useful as it allows us to be inclusive of processes that operate over distinct transition times. We would prefer to retain inclusivity.

As noted above, we have included a new paragraph at the top of the paper to discuss this.

“Forming a universal definition of Antarctic EEEs, which fits all circumstances, is challenging. Antarctica has an intrinsically extreme environment, which is already subject to magnified change relative to that seen in the rest of the planet due to fossil fuel burning and resultant global heating. On top of this backdrop, incidents occur that can take the form of short shocks, irreversible step changes or persistent changes acting across many different aspects or features of the environment. We offer an inclusive approach to EEEs, by considering the effects of such incidents, which can be individual, collective or cascading, and which remain sudden and/or unexpected compared to what has traditionally been assumed to occur in Antarctica.”

The second major issue is geographic imbalance. Most operational examples are drawn from BAS-linked facilities, with McMurdo and Wilkins Aerodrome serving only as limited counterpoints. Major national programmes such as Concordia, Neumayer, Syowa, Jang Bogo, and Zhongshan are absent. This is particularly noticeable because Section 4 raises China’s growing Antarctic research role while not incorporating Chinese operational experience into the broader discussion. The authors should either broaden the evidence base or explain why the current examples are sufficiently representative.

This is a good point and one that stems from our experiences and knowledge. We now make this point explicitly. “Here, based on our experiences and knowledge (primarily with, but not restricted to, UK operations), we discuss how EEEs will lead to impacts...”.

The mitigation discussion also needs further development. The abstract promises discussion of how impacts can be mitigated, but this remains underdeveloped in the main text. Autonomous underwater vehicles are mentioned in the abstract but not meaningfully discussed in the body, and land-based autonomous platforms are effectively absent. More attention should also be given to the practical constraints on adaptation, including cost, infrastructure hardening, and logistical redundancy.

We have added a sentence when we consider drones and AUVs as follows: “Use of remotely operated and autonomous devices is likely to become an increasingly major theme of investigation.” We have also added a new topic of proposed investigation for IPY5 “and (7) how to mitigate their impact on science using drones and other autonomous systems.”

Finally, Section 4 raises an interesting governance point but does not fully develop it. The observation that China has overtaken the USA in Antarctic research output could have important implications, but these are only briefly noted and not analysed in depth. If this argument is retained, its implications for burden-sharing, infrastructure dependence, and Antarctic governance should be discussed more directly.

We are grateful to the review for flagging this point. The following text has been added to Section 4:

“In a continent devoted to peace and science and where the demonstration of substantial scientific research activity is necessary to participate in Antarctic governance, and with 58 signatories to the Treaty, a shift in scientific activity and change in levels of academic output may have broad implications [58]. These may include an increasing role by some Parties in the provision of research to respond to the information and knowledge needs of the bodies of the Antarctic Treaty system (such as the Antarctic Treaty Consultative Meeting (ATCM), Committee for Environmental Protection (CEP) or Commission for the Conservation of Antarctic Marine Living Resources (CCAMLR)), as well as a changing relationship between Parties operating key infrastructure, such as airstrips, and those seeking to use these facilities to deliver their field research activities.”

Minor concerns:

- There are also some reference and presentation issues that should be corrected. The EPA ice pier reference appears to need renumbering.

Many thanks for this comment. We have edited the reference accordingly.

- Reference [51], a The Conversation article, is not a sufficiently strong source for the claim about China’s leadership in Antarctic research output; a peer-reviewed bibliometric source would be more appropriate.

We don’t think the issue is in doubt, but we could use the following citations as a replacement and would be happy to receive editorial advice.

“Larson, K., Aksnes, D. W., Danell, R., King, M., Leane, E., & Olofsson, I. (2025). The Antarctic Research Trends Report 2025. UArctic & Umeå university. <https://doi.org/10.5281/zenodo.18402791> “

and/or

“Zherebchuk, S., Kudas, D., & Kuz, S. (2025). The role of research in Antarctic diplomacy: A scientometric analysis of the Treaty System's impact. *Ukrainian Antarctic Journal*, 23(1 (30)), 100-119.”

- The 2018 COMNAP survey data cited in Section 2.1 are now dated and should be updated if more recent figures are available.

Unfortunately, the COMNAP survey has not been repeated. However, we do highlight the recent provision of information by COMNAP on actions taken by national Antarctic programmes to address the impacts of a changing climate.

- The abstract and body should also be aligned more carefully with respect to mitigation using autonomous platforms.

We have tried to do this without overloading the text on this aspect in ways discussed above.

Response to referees' comments

"Antarctic science operations must account for climate change and extreme environmental events"

Dear Nicola,

Many thanks for agreeing in principle to publish the above paper. The one outstanding item was to "clearly articulate limitations in representativeness, particularly as they relate to Antarctic programmes run by other countries, for example, China".

The term COMNAP use for programmes run by other countries is 'governmental National Antarctic Programs'. We have included this terminology and introduced a new final sentence to the Introduction for it to read as follows:

"Here, based on our experiences and knowledge (primarily with, but not restricted to, UK operations), we discuss how EEEs will lead to impacts on science and logistics in Antarctica and the Southern Ocean, and how – if at all – such impacts can be mitigated. We consider how scientific experimentation may be adversely affected through reassessments of risk, operations and policy. **We note that there is a broad range of governmental National Antarctic Programs, which will each need to determine how EEEs affect activities and plans in ways highlighted by, but not restricted to, our considerations.**"

We hope this change is sufficient to cover the point made by the referee.

We have completed the form as requested and believe we have included everything you require at this point.

Kind regards,

Martin Siegert